# A longitudinal study of the pulmonary mycobiome in subjects with and without chronic obstructive pulmonary disease

**Einar M. H. Martinsen**[1]*, **Tomas M. L. Eagan**[1,2], **Harald G. Wiker**[1,3], **Elise O. Leiten**[1], **Gunnar R. Husebø**[1,2], **Kristel S. Knudsen**[2], **Solveig Tangedal**[1,2], **Walter Sanseverino**[4], **Andreu Paytuví-Gallart**[4], **Rune Nielsen**[1,2]

**1** Department of Clinical Science, University of Bergen, Bergen, Norway, **2** Department of Thoracic Medicine, Haukeland University Hospital, Bergen, Norway, **3** Department of Microbiology, Haukeland University Hospital, Bergen, Norway, **4** Sequentia Biotech SL, Barcelona, Spain

* einar.martinsen@uib.no

## Abstract

### Background

Few studies have examined the stability of the pulmonary mycobiome. We report longitudinal changes in the oral and pulmonary mycobiome of participants with and without COPD in a large-scale bronchoscopy study (MicroCOPD).

### Methods

Repeated sampling was performed in 30 participants with and 21 without COPD. We collected an oral wash (OW) and a bronchoalveolar lavage (BAL) sample from each participant at two time points. The internal transcribed spacer 1 region of the ribosomal RNA gene cluster was PCR amplified and sequenced on an Illumina HiSeq sequencer. Differences in taxonomy, alpha diversity, and beta diversity between the two time points were compared, and we examined the effect of intercurrent antibiotic use.

### Results

Sample pairs were dominated by *Candida*. We observed less stability in the pulmonary taxonomy compared to the oral taxonomy, additionally emphasised by a higher Yue-Clayton measure in BAL compared to OW (0.69 vs 0.22). No apparent effect was visually seen on taxonomy from intercurrent antibiotic use or participant category. We found no systematic variation in alpha diversity by time either in BAL (p-value 0.16) or in OW (p-value 0.97), and no obvious clusters on bronchoscopy number in PCoA plots. Pairwise distance analyses showed that OW samples from repeated sampling appeared more stable compared to BAL samples using the Bray-Curtis distance metric (p-value 0.0012), but not for Jaccard.

### Conclusion

Results from the current study propose that the pulmonary mycobiome is less stable than the oral mycobiome, and neither COPD diagnosis nor intercurrent antibiotic use seemed to influence the stability.

**Data Availability Statement:** The dataset and code supporting the conclusions of this article is available in the DRYAD repository. Age and sex are omitted from the metadata due to privacy

concerns. Available from: https://doi.org/10.5061/dryad.kd51c5b41.

**Funding:** The MicroCOPD study was funded by unrestricted grants and fellowships from Helse Vest, GlaxoSmithKline, Bergen Medical Research Foundation, and the Endowment of Timber Merchant A. Delphin and Wife through the Norwegian Medical Association. No specific grant or award numbers exist for the specific funding. The MicroCOPD funders had no role in study design, data collection and analysis, decision to publish, or preparation of the manuscript. Sequentia Biotech SL provided support in the form of salaries for authors Walter Sanseverino and Andreu Paytuví-Gallart, but not study design or data collection. All funding of data collection and laboratory analyses were from the MicroCOPD Study. Walter Sanseverino and Andreu Paytuví-Gallart both contributed to interpretation of results, and preparation of the manuscript. The specific roles of all authors are articulated in the 'author contributions' section.

**Competing interests:** I have read the journal's policy and the authors of this manuscript have the following competing interests: Einar M. H. Martinsen, Elise O. Leiten, Gunnar Husebø, and Solveig Tangedal declare no conflict of interest. Walter Sanseverino and Andreu Paytuví-Gallart are employed at Sequentia Biotech SL. Tomas M. L. Eagan reports lecture fees from Boehringer and AstraZeneca, and grants from GlaxoSmithKline outside the submitted work. Harald G. Wiker reports being head of the educational programme for medicine at the University of Bergen. Kristel S. Knudsen reports lecture fees from Boehringer Ingelheim and Roche. Rune Nielsen reports grants from Boehringer Ingelheim, GlaxoSmithKline, AstraZeneca, and the Timber Merchant Delphins Endowment, in addition to lecture fees from GlaxoSmithKline and AstraZeneca. Rune Nielsen also reports being unpaid member of The Norwegian Respiratory Society, unpaid GOLD national delegate, unpaid ERS national delegate, and paid member of the Reference group for new Norwegian guidelines for COPD (The Norwegian Directorate of Health). This does not alter our adherence to PLOS ONE policies on sharing data and materials.

## Introduction

The fungal part of the pulmonary microbiome, the mycobiome, has gained more interest recent years [1]. An important part of investigating the potential presence of a pulmonary mycobiome, is to examine the temporal variability upon repeated examinations.

To our knowledge, no study has published data on the stability of the pulmonary mycobiome in subjects without lung disease. Further, we are only aware of four mycological studies where repeated measurements were performed in a chronic obstructive pulmonary disease (COPD) population [2–5]. The first study collected two sputum samples in stable COPD patients, and observed low repeatability of *Aspergillus fumigatus* cultures [2]. Culture-based techniques have low sensitivity [6], and often fail to identify the complete mycobiome [7]. It is thus hard to generalise results from the culture-based study [2]. The three other studies examined longitudinal changes in the lung mycobiome during exacerbations [3–5]. No significant changes were seen in airway mycobiome profiles, alpha diversity, or beta diversity, despite treatment with oral antibiotics and corticosteroids in the study by Tiew et al. [3]. No statistics were applied on the repeated samples in Liu et al.'s study [5], while Su et al. found an unstable mycobiome in included patients [4]. A few studies have performed longitudinal analysis of sputum samples in cystic fibrosis (CF) [8–10]. However, sputum samples are susceptible to contamination from the upper respiratory tract and oral cavity, and may not be representative of the pulmonary mycobiome. Oropharyngeal contamination can be minimised by protected bronchoscopic sampling [11], which in addition provides accurate assessment over the actual sample site.

The Bergen COPD Microbiome study (short name "MicroCOPD") is a large single-centre study of the pulmonary microbiome conducted in Bergen, Norway [12]. Samples were collected from both the upper and the lower airways, the latter providing particular attention to contamination by using a sterile inner catheter in the working channel of a bronchoscope. We have previously published analyses on the oral and pulmonary mycobiome, in addition to comparisons between the healthy and the COPD oral and pulmonary mycobiomes using data from the MicroCOPD study [13]. In the current paper, we report on the stability of the oral and pulmonary mycobiomes in participants with and without COPD who underwent bronchoscopies at two different time points, and the potential effects on the mycobiomes from intercurrent antibiotic use.

## Materials and methods

### Study design and population

The design and data collection in the MicroCOPD study is previously published [12]. Briefly, the study was carried out at the outpatient clinic of the Department of Thoracic Medicine, Haukeland University Hospital. Participant enrolment started April 11[th], 2013, and finished June 5[th], 2015. A total of 93 COPD patients and 100 participants without lung disease (controls) provided samples for mycobiome analyses. Controls were mainly recruited among participants from a previous general population study [14], in a non-random manner. Participants were excluded if they were hypoxemic despite supplemental oxygen, hypercapnic, had an increased bleeding risk, or possessed certain cardiac risk factors like a mechanical valve. Ongoing COPD exacerbations and recent use of antibiotics or systemic corticosteroids were reasons for postponing participation in order to examine the stable mycobiome [12]. The study was conducted in accordance with the declaration of Helsinki and guidelines for good clinical practice. The regional committee of medical ethics Norway north division (REK-NORD) approved the project (project number 2011/1307), and all participants provided informed written consent.

Among the 193 participants, 30 participants with COPD and 21 controls were eligible for a repeated bronchoscopy 3–12 months after the first. Inclusion for a second bronchoscopy stopped when the main study ended. Information on treatment received outside of the study in the time between the two study visits was collected. Antibiotic use in this period was termed intercurrent antibiotic use and was not part of the exclusion criteria unless given recently before the bronchoscopy.

COPD was identified by a post-bronchodilator $FEV_1/FVC$ ratio $< 0.7$ and a corresponding symptom history, in accordance with the Global Initiative for Chronic Obstructive Lung Disease (GOLD) report [15]. The COPD diagnoses were carefully re-evaluated by experienced pulmonologists in late 2015 based on spirometry, radiologic imaging, respiratory symptoms, and disease history.

### Data collection

At the day of bronchoscopy, peripheral venous blood was drawn, and post-bronchodilator spirometry was performed. Information on comorbidities, medication use, demographics, smoking habits, and respiratory symptoms was collected in structured interviews.

Participants provided an oral wash sample (OW) immediately prior to the bronchoscopy procedure by gargling 10 ml of fluid from a bottle of phosphate buffered saline (PBS) for 1 minute. PBS from the same bottle was used for all samples from each unique participant, including OW, a negative control sample (NCS), and bronchoalveolar lavage (BAL). The NCS was collected from the PBS bottle directly into a sterile container and was thereafter subject to the same laboratory protocol as the procedural samples. Bronchoscopy was performed with the study subject in supine position and via oral access. Participants were offered light sedation with alfentanil. Topical anaesthesia was delivered by 10 mg/dose oral lidocaine pre-procedurally, and during the procedure through a spray catheter. Two fractions of 50 ml protected BAL were collected using a wax-tip protected catheter (Plastimed Combicath, prod number 58229.19) from the right middle lobe. The second portion of the BAL was utilised for mycobiome analysis in the current study. At the repeated bronchoscopy, a detailed medical history including medication use and exacerbation history since the previous bronchoscopy was obtained.

### DNA extraction and sequencing

A combination of enzymatic lysis with lysozyme, mutanolysin and lysostaphin, and mechanical lysis methods using the FastPrep-24 as described by the manufacturers of the FastDNA Spin Kit (MP Biomedicals, LLC, Solon, OH, USA) was used to extract DNA. Samples were prepared for sequencing of the internal transcribed spacer (ITS) 1 region using a modified version of the Illumina 16S Metagenomic Sequencing Library Preparation guide (Part no. 15044223 Rev. B). The PCR amplification included 28 cycles using primer set ITS1-30F/ITS1-217R, where the sequences are `GTCCCTGCCCTTTGTACACA` and `TTTCGCTGCGTTCTTCATCG` [16], followed by an index PCR with 9 cycles. DNA sequencing was performed using paired-end sequencing (2x250 cycles) on an Illumina HiSeq sequencer (Illumina Inc., San Diego, CA, USA). Samples were sequenced in two different sequencing runs, and OW, BAL, and NCS from a given participant were always included in the same sequencing run. Both sequencing runs included samples from controls and participants with COPD.

### Bioinformatic processing

Bioinformatic processing and analyses were performed with QIIME 2 version 2019.1 and 2019.10 [17], and R version 4.0.3 [18]. Raw sequence data was quality controlled (via

q2-demux) and trimmed using the q2-itsxpress plugin [19]. The Divisive Amplicon Denoising Algorithm version 2 (DADA2) (via q2-dada2) was used to identify and remove low-quality reads and chimeric sequences, resulting in the creation of exact amplicon sequence variants (ASVs) [20]. These ASVs were further curated using the LULU R package to remove artefactual ASVs [21]. Singletons, i.e. ASVs found in only one sample, and ASVs with a total abundance of less than 10 sequence reads across all samples, were excluded. The Decontam R package was used to identify ASVs likely to be contaminants using the prevalence-based method (user defined threshold = 0.5) [22], and the contaminants were subsequently removed. Taxonomy was assigned to ASVs using the q2-feature-classifier [23] classify-sklearn [24] with a UNITE database for fungi with clustering at 99% threshold level [25, 26]. Resulting ASVs assigned only as *Fungi* at kingdom level were manually investigated using the Nucleotide Basic Local Alignment Search Tool (BLASTN) program maintained by the National Center for Biotechnology Information (NCBI) in the Nucleotide database [27, 28]. ASVs with unambiguous BLASTN results were repeatedly assigned to new taxonomic assignments using UNITE databases with fungi or all eukaryotes with different threshold levels [25, 29–31] (via q2-feature-classifier [23] classify-sklearn [24] and classify-consensus-blast [32]). They were included for further analyses if the new assignments matched the BLASTN result. ASVs with ambiguous or non-fungal BLASTN results were discarded. Alpha diversity metrics (Shannon index) and beta diversity metrics (Bray-Curtis dissimilarity and Jaccard similarity coefficient) were estimated post rarefaction (subsampling without replacement) at sampling depth 2000 (via q2-diversity). The sampling depth was set as high as possible while excluding a minimum of samples.

## Statistical analyses

Demographics were evaluated as means with standard deviations or percentages using Stata version 16 [33]. A flow chart showing the bioinformatic processing was created with the Flowchart Designer version 3 (http://flowchart.lofter.com). Taxonomy was presented as relative abundances per individual on genus level grouped by participant group and intercurrent antibiotic use. We calculated the Yue-Clayton measure of dissimilarity ($1-\theta_{YC}$) between each sample pair from the first and the second bronchoscopy [34]. Medians of Shannon indexes from the first and second bronchoscopy were compared with Wilcoxon signed-rank test for paired analysis stratified by sample type. Differences in beta diversity were presented as principal coordinates analysis (PcoA) plots, and Procrustes analyses were performed including the three first axes from the PcoA result to check for differences in beta diversity between repeated bronchoscopies using the Vegan package in R [35]. Pairwise distances between the first and the second bronchoscopy were calculated using q2-longitudinal [36] and separated on sample type compared with Kruskal-Wallis test, and on time between procedures.

## Results

### Flow chart

An overview of the data processing is shown in Fig 1.

### Demographics of participants

Participants with COPD had a higher smoking exposure measured in pack years, more comorbidities, and a lower $FEV_1$ compared to the controls (Table 1). These differences were not tested statistically as they were expected.

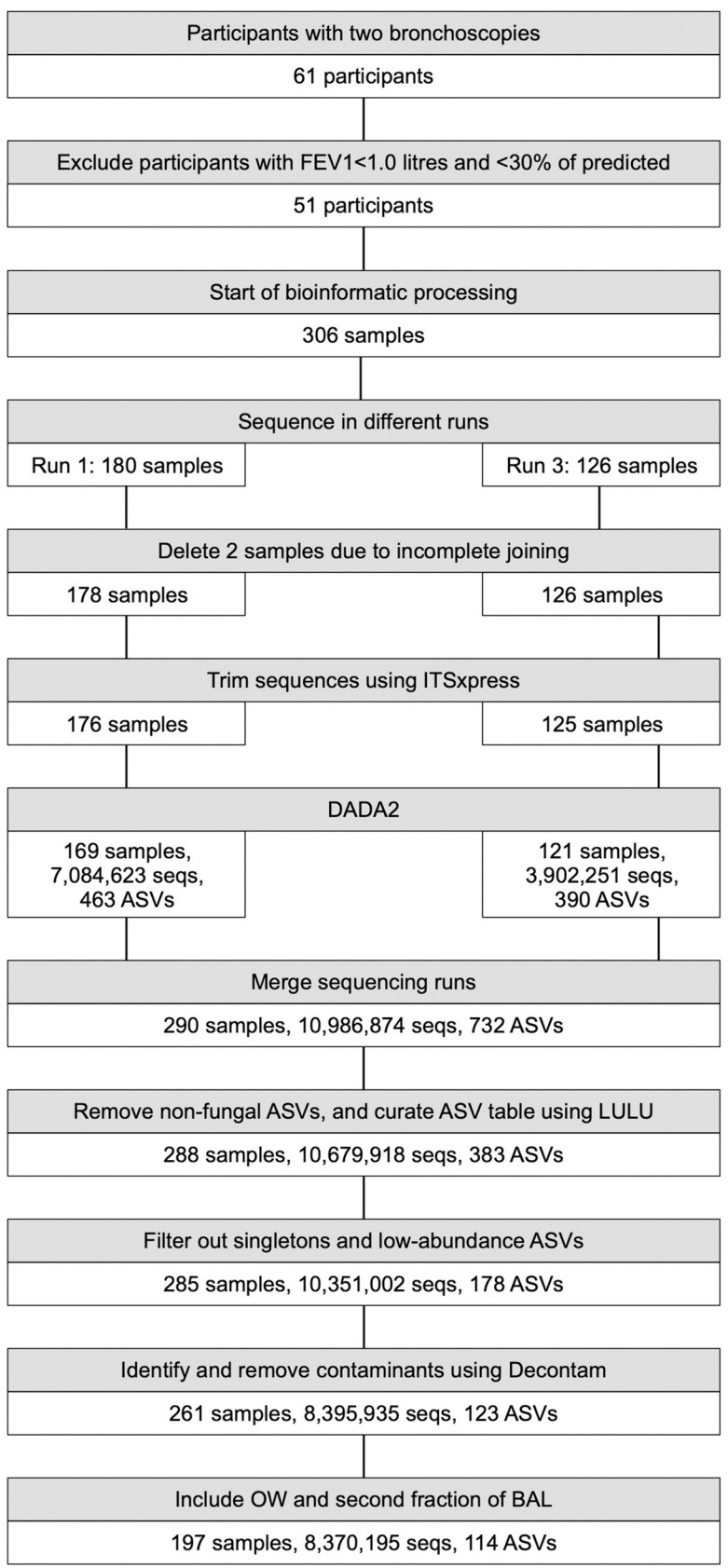

**Fig 1. Flow chart of fungal longitudinal analyses in the MicroCOPD study.** DADA2: Divisive Amplicon Denoising Algorithm version 2, seqs: sequences, ASV: amplicon sequence variant, OW: oral wash, BAL: bronchoalveolar lavage. Samples were sequenced in three different runs before trimming and denoising. Data from different sequencing runs were merged, and then processed to exclude contaminants and negative control samples prior to analyses.

## Taxonomy

Taxonomy of OW sample pairs is plotted in Fig 2A and S1 Fig. The nine most abundant taxa were visualised individually, while taxa with low abundances were merged and visualised as *Others*. More details are given in S2A Fig, showing all taxa, as well as Yue-Clayton measures. The Yue-Clayton measure is 0 with perfect similarity and 1 with perfect dissimilarity. Taxonomies from the majority of OW sample pairs seemed unchanged between the bronchoscopies, but it appeared that sample pairs without *Candida* dominance showed more instability (i.e. participant 16, 24, 35, 39, and 50 in S2A Fig). The average Yue-Clayton measure from OW sample pairs was 0.22.

Taxonomy of 45 BAL sample pairs is similarly plotted in Fig 2B and S2B Fig. Only sample pairs with BAL samples from both bronchoscopies were included, which explains the lower number of sample pairs compared to OW. *Candida* is still the most dominant taxon, but less than for OW samples. The average Yue-Clayton measure from BAL sample pairs was 0.69. Few differences were visually seen between participant categories in both sample types.

## Diversity

A considerable number of our BAL samples were omitted from diversity analyses due to sub-sampling (rarefaction). There was no significant difference in alpha diversity measured by

**Table 1. Demographics of participants included in fungal longitudinal analyses in the MicroCOPD study.**

| Variable | Control, n = 21 | COPD, n = 30 | Comparison, p-value |
|---|---|---|---|
| **Age (SD)** | 66.8 (5.5) | 68.7 (6.2) | 0.27 |
| **Male sex (%)** | 14 (66.7) | 19 (63.3) | 1.0 |
| **Intercurrent antibiotic use (%)** | 4 (19.1) | 6 (20) | 0.93 |
| **Number of medications (SD)** | 1.6 (2.0) | 5.6 (3.5) | < 0.01 |
| **Use of inhaled steroids (%)** | - | 18 (60) | NA |
| **Number of comorbidities (SD)** | 0.9 (1.3) | 1.5 (1.4) | 0.06 |
| **$FEV_1$, % of predicted (SD)** | 104.3 (12.4) | 59.7 (16.0) | < 0.01 |
| **FVC, % of predicted (SD)** | 116.0 (13.8) | 100.0 (15.8) | < 0.01 |
| **$FEV_1$/FVC-ratio (SD)** | 0.71 (0.05) | 0.47 (0.11) | < 0.01 |
| **Pack years (SD)** | 22.1 (12.5) | 32.1 (14.9) | 0.02 |
| **Smoking status (%)** | | | 0.25 |
| Daily | 7 (33.3) | 6 (20) | |
| Ex | 13 (61.9) | 24 (80) | |
| Never | 1 (4.8) | - | |
| **mMRC Grade 2 and higher (%)** [*] | | | NA |
| Grade 2 level ground | - | 7 (24.1) | |
| Grade 3 100 meters | - | 5 (17.2) | |
| Grade 4 resting dyspnoea | - | - | |
| **Time between procedures in days (SD)** | 156 (33.7) | 151 (62.7) | 0.22 |

$FEV_1$: forced expiratory volume in 1 second, FVC: forced vital capacity, mMRC: modified medical research council dyspnoea scale. Data from first exam were used, except for sex, intercurrent antibiotic use, and time between procedures. Comparisons were done using t-test or Kruskal-Wallis depending on normality for continuous variables and chi-square test or Fisher's exact test for categorical variables.

[*] One participant with COPD missed information on mMRC.

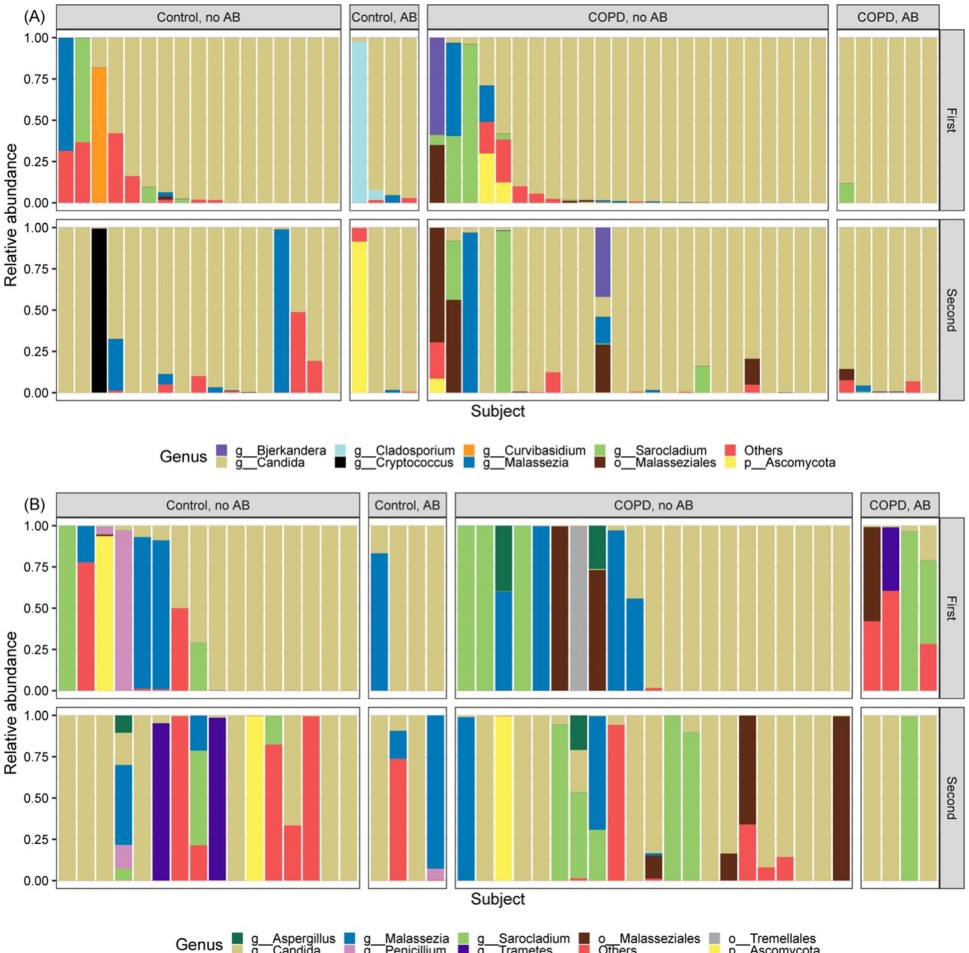

**Fig 2.** Most abundant fungal taxonomic assignments in (A) oral wash (51 participants) and (B) bronchoalveolar lavage (45 participants). AB: Intercurrent antibiotic use, First: first bronchoscopy, Second: second bronchoscopy. Taxa are sorted on *Candida* in the first bronchoscopy. Each column represents a sample, and columns from the first bronchoscopy and the second bronchoscopy correspond to each other. That means, a first bronchoscopy column above and the corresponding second bronchoscopy column below show samples from the same participant. Not all sequences could be assigned taxonomy at the genus level and are therefore displayed as *o__Malasseziales*, *o__Tremellales*, or *p__Ascomycota*.

Shannon index between the first and the second bronchoscopy in OW or BAL (Fig 3). Changes in Shannon index seemed unpredictable, with some being stable, some increasing, and some decreasing. The one participant with intercurrent antibiotic use and BAL available for diversity analyses showed a distinct increase in Shannon index. Similar observations were not seen in the OW group, where samples from participants with intercurrent antibiotic use both increased and decreased in alpha diversity.

PCoA plots were created to visualise differences in beta diversity (S3–S8 Figs). There were no obvious clusters on bronchoscopy number plotted with OW and BAL combined (S3 Fig) or for each sample type separately (S5 and S7 Figs). The $M^2$ from Procrustes transformations were above 0.3 for all transformations (S6 and S8 Figs), which is often interpreted as dissimilar. The $M^2$ statistic is a measure of fit, which increases with less concordance. However, results were only significant for OW samples (S6 Fig. Bray-Curtis: $M^2 = 0.314$, $p = 0.001$, Jaccard: $M^2 = 0.7987$, $p = 0.002$). The pairwise distance between each subject pair differed between OW

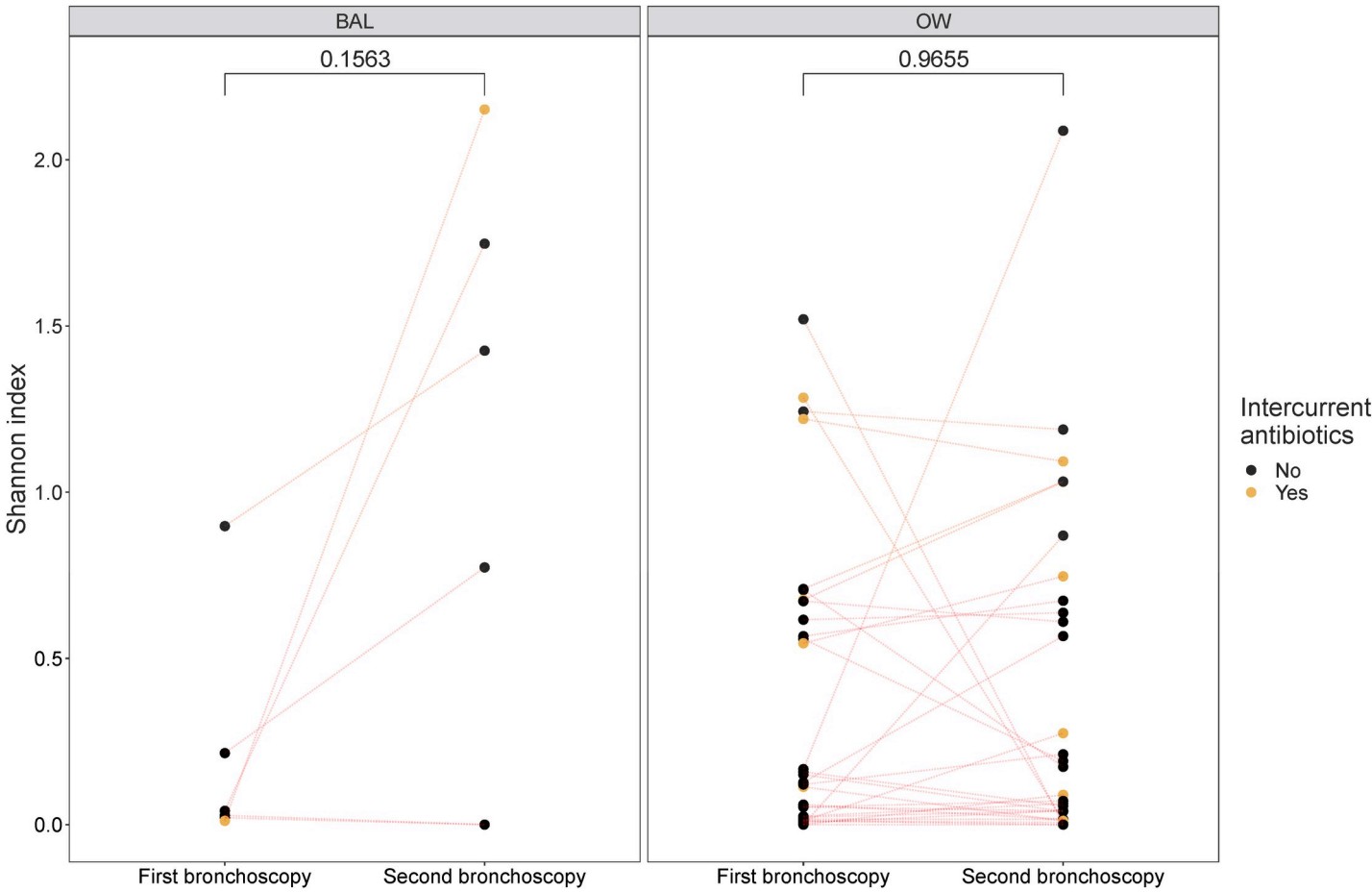

**Fig 3. Analysis of alpha diversity differences between first and second bronchoscopy.** BAL: bronchoalveolar lavage, OW: oral wash. A line was drawn between samples from the same participant, divided in BAL (12 samples) and OW (62 samples). Differences in Shannon were statistically tested with Wilcoxon signed-rank test with a significance level of 0.05.

and BAL using Bray-Curtis (Kruskal-Wallis chi-squared = 10.466, p-value = 0.001216), but not for Jaccard (Fig 4A). No specific trends were seen with increasing time between the bronchoscopies (Fig 4B).

## Discussion

We have reported the oral and pulmonary mycobiome in controls and participants with COPD sampled at two time points. The oral mycobiome showed a higher degree of stability compared to the pulmonary mycobiome. Neither intercurrent antibiotic use nor time between bronchoscopies seemed to influence the mycobiomes.

Two CF studies have explored the stability of the pulmonary mycobiome using targeted amplicon sequencing. In the first paper, relative abundances of fungal taxa appeared to change by time, but were not statistically analysed [8]. The other CF study observed high relative abundances of *Candida* species which did not change during antibiotic treatment [10]. Low sample sizes, lack of controls, and frequent use of antibiotics and steroids, make comparisons of findings from the CF studies to our results difficult. Krause et al. observed a stable mycobiome in repeated BAL performed only 4–7 days after baseline [37]. However, results were based on 4 intubated and mechanically ventilated patients with pneumonia. A longitudinal

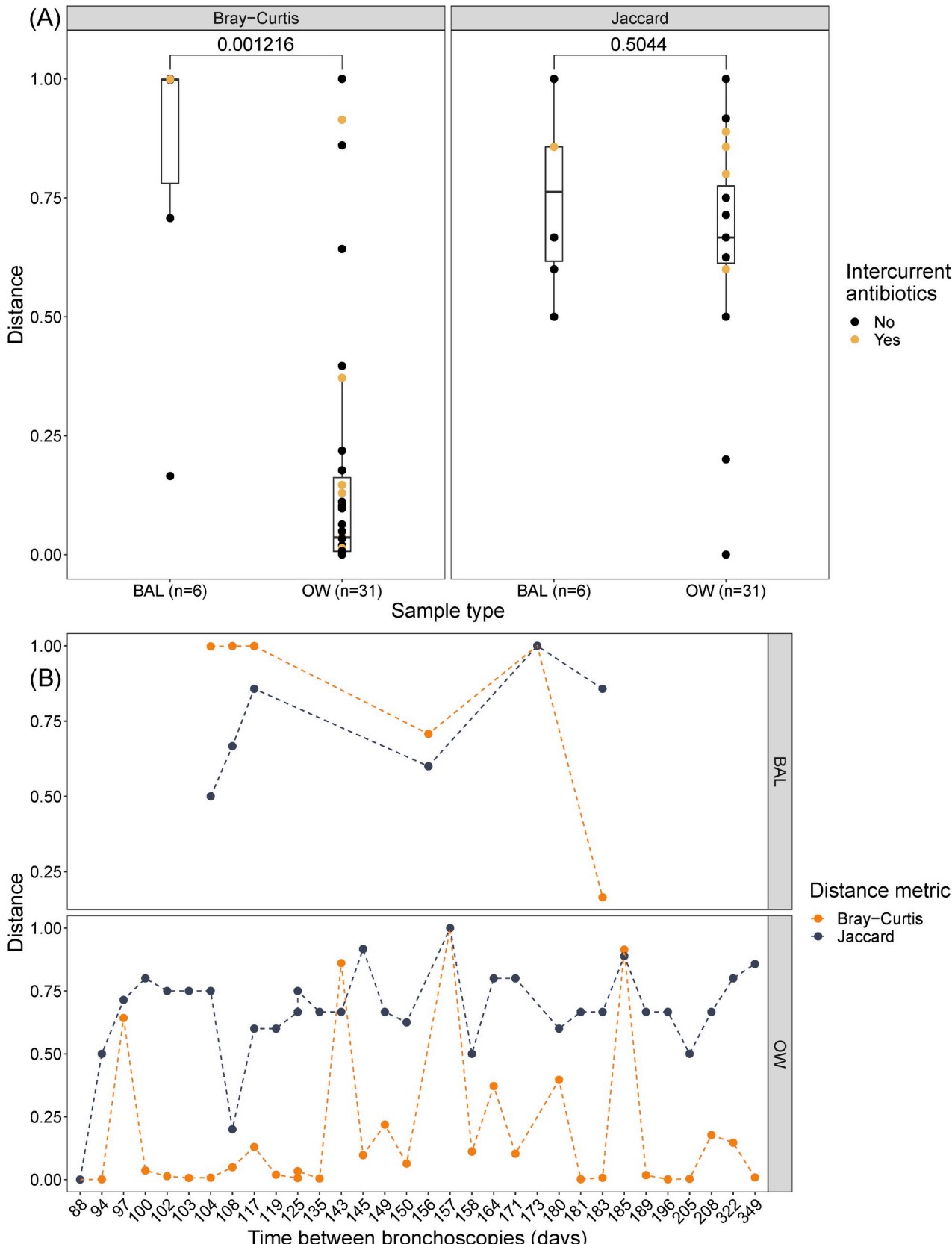

**Fig 4.** Analysis of pairwise distances between first and second bronchoscopy grouped on (A) sample type and (B) time. BAL: bronchoalveolar lavage, OW: oral wash. Differences in pairwise distances from the first and the second bronchoscopy were statistically tested with Kruskal-Wallis test with a significance level of 0.05. Time was measured in days between the two bronchoscopy procedures.

study of filamentous fungi in the airways of participants with COPD and controls was performed by Bafadhel et al. by collecting sputum samples at baseline and three months later [2]. They found a poor repeatability of *Aspergillus fumigatus* in the participants with COPD, but the study focused exclusively on *A. fumigatus* and included cultures only. Su et al. reported an unstable mycobiome during exacerbations [4]. The results were based on sputum samples from six patients with COPD, and sampling was only performed during the hospital stay. Liu et al. examined the pulmonary mycobiome during an acute exacerbation in four participants with COPD, but with emphasis on the difference between sputum and oropharyngeal swabs [5]. No statistics were applied to the results from consecutive collected samples in the papers by Su and Liu et al. [4, 5], thus limiting interpretation. Tiew et al. collected sputum samples from 34 participants with COPD before an exacerbation, within 24 hours of an acute exacerbation, and again two weeks post exacerbation following treatment [3]. No significant changes were seen in airway mycobiome profiles, alpha diversity, or beta diversity, suggesting that treatment for an acute exacerbation did not influence the lung mycobiome. However, the sampling interval was short, and taxonomy was presented at group level, which could have masked intraindividual changes. The discrepancy to the taxonomic instability observed in our BAL samples, could perhaps be attributed to the use of sputum samples in the study by Tiew et al. [3]. Sputum samples are more prone to oral contamination, and we indeed observed a higher stability in our oral samples compared to BAL. Bronchoscopy and BAL have been associated with a potential immune response manifested by post-procedural fever and flu-like symptoms [38, 39]. We speculate that the participants experiencing an immune response could acquire immunisation against their own respiratory microorganisms during bronchoscopy, thereby eliminating these microorganisms and creating a taxonomic shift. The oral cavity is less triggered by the bronchoscopy, and is more accustomed to exposures, which could explain higher stability in OW samples. Influence on the lung mycobiome from bronchoscopy due to immunisation could be an important finding, but needs to be examined with multiple sampling time points in larger populations to be confirmed and to reveal any potential clinical implications.

In the current analyses, most sample pairs had a taxonomy dominated by *Candida*, particularly for OW samples. No apparent effect was visually seen on the mycobiomes from intercurrent antibiotic use or participant category. The lower average Yue-Clayton observed in OW sample pairs compared to BAL (0.22 vs 0.69) suggests a higher degree of stability in OW samples. Visually, it seemed as most OW pairs with a *Candida* dominance in the first sampling had *Candida* dominance in the second sampling, whereas BAL bronchoscopy pairs with a *Candida* dominance in the first bronchoscopy more often lacked *Candida* in the second bronchoscopy. By looking at the taxonomy, one gets the impression that samples without *Candida* experience higher fluctuations, suggesting *Candida* is involved in maintaining stability. In congruence with our results, the *Candida* rich oral mycobiome has previously been shown to have high stability [40]. In contrast to the stability in the oral mycobiome, analyses of the gut mycobiome of the Human Microbiome Project found high intraindividual variability, but *Saccharomyces*, *Malassezia*, and *Candida* were commonly identified longitudinally within individuals [41].

In the current paper, we observed that fungal alpha diversity (i.e. within-sample diversity) measured by Shannon index varied within individuals by time, but it was stable at group level for both OW and BAL samples. No consistent effect on each participant's temporal change in

alpha diversity was visually seen from intercurrent antibiotic use, but numbers were too small to analyse statistically. Subsampling by rarefaction was done to account for different sampling depth among samples. Rarefaction is included in QIIME 2's diversity metric plugin, and it resulted in a considerable omission of BAL samples. Diversity analyses were thus performed on controls and participants with COPD together but stratified on sample type. This was considered appropriate due to small observed differences in taxonomies from the participant categories. Additionally, in a previous paper, we reported few differences between the healthy and COPD mycobiomes in the MicroCOPD study [13]. Distance matrices used in beta diversity (i.e. between-sample diversity) analyses are multidimensional. PCoA ordination is frequently used to display matrices at a low dimensional space, and to examine dissimilarities in composition. By applying a Procrustes transformation, one can examine how similar different coordinate matrices with corresponding points are. PCoA plots in the current paper showed no clusters on repeated procedures nor on antibiotic use, but the Procrustes transformation indicated a dissimilarity between first and second bronchoscopies in OW samples. In contrast, pairwise distances from repeated bronchoscopies drawn from beta diversity distance matrices were significantly higher for OW samples compared to BAL samples, again suggesting higher stability in OW samples. Significant difference was only observed in Bray-Curtis, and not for Jaccard. High relative abundances of *Candida* in OW samples could impact the abundance-based Bray-Curtis metric more than the absence/presence based Jaccard metric, and thus yield low pairwise distances in OW. Lack of any specific trend with increasing time between the bronchoscopies suggests that the mycobiomes are equally similar or dissimilar by time. Previous studies on temporal diversity changes in the pulmonary mycobiome have been conflicting, with some reporting stability [3, 8, 10] and some reporting fluctuations [4, 9]. Comparisons to our study are difficult due to inclusion of sputum from CF patients [8–10] or COPD exacerbations [3, 4].

The current study is, to our knowledge, the first to analyse the stability of the pulmonary mycobiome of participants with COPD and healthy controls by use of targeted amplicon sequencing. Nevertheless, some limitations should be mentioned. First, participants underwent bronchoscopy with different time intervals. Lack of a defined time interval might introduce biases, but pairwise distances drawn from Bray-Curtis and Jaccard distance matrices did not change much by time. Furthermore, when we sorted samples in the taxonomic barplots on time between bronchoscopies, no specific pattern was observed (S1 Fig). Second, possible batch effects occur when multiple sequencing runs are included. Batch effects have been considered on the whole MicroCOPD mycobiome data set, suggesting some effect on beta diversity and *Sarocladium* abundance [13]. Third, despite being a large study, some analyses suffer from low statistical power. Fourth, including only the second part of the BAL could impact the results. Another paper by our research group has shown high similarity between the first and the second fraction of BAL, suggesting little impact from choice of fraction [11]. Finally, it has been suggested that OW and BAL samples suffer from low reproducibility when repeatedly extracted and sequenced, but reproducibility improved when contamination was accounted for [42]. We used protected sampling methods and contamination removal through the validated R package Decontam, thus potentially reducing reproducibility issues.

Frequent use of cross-sectional study designs in mycobiome studies prevent conclusions on the mycobiome's stability. Results from the current study propose that the pulmonary mycobiome is less stable than the oral mycobiome, and neither COPD diagnosis nor intercurrent antibiotic use seemed to influence the stability. Longitudinal studies with well-defined time intervals are needed to assess whether the pulmonary mycobiome is defined by large fluctuations or core taxa.

## Supporting information

**S1 Fig.** Most abundant fungal taxonomic assignments in (A) oral wash and (B) bronchoalveolar lavage. AB: Intercurrent antibiotic use, First: first bronchoscopy, Second: second bronchoscopy. Samples are sorted on time between bronchoscopies. Each column represents a sample, and columns from the first bronchoscopy and the second bronchoscopy correspond to each other. That means, a first bronchoscopy column above and the corresponding second bronchoscopy column below show samples from the same participant. Not all sequences could be assigned taxonomy at the genus level and are therefore displayed as o__Malasseziales, o__Tremellales, or p__Ascomycota.
(PDF)

**S2 Fig.** Yue-Clayton plots of (A) oral wash and (B) bronchoalveolar lavage sample pairs from participants examined twice in the MicroCOPD study. YC: Yue-Clayton measure, First: first bronchoscopy, Second: Second bronchoscopy. A Yue-Clayton measure of 0 means identical sample pairs, while a Yue-Clayton measure of 1 means unidentical sample pairs.
(PDF)

**S3 Fig.** Principal coordinates analysis plot using (A) Bray-Curtis and (B) Jaccard coloured by bronchoscopy number and sample type. First BAL: bronchoalveolar lavage samples from first bronchoscopy, First OW: oral wash samples from first bronchoscopy, Second BAL: bronchoalveolar lavage samples from second bronchoscopy, Second OW: oral wash samples from second bronchoscopy.
(PDF)

**S4 Fig.** Principal coordinates analysis plot using (A and B) Bray-Curtis and (C and D) Jaccard coloured by intercurrent antibiotic use and sample type. BAL: bronchoalveolar, OW: oral wash.
(PDF)

**S5 Fig.** Principal coordinates analysis plot of OW samples using (A and C) Bray-Curtis and (B and D) Jaccard coloured by bronchoscopy number and intercurrent antibiotic use with and without lines. First: first bronchoscopy, Second: Second bronchoscopy. A line was drawn between samples from the same participant. Red lines refer to participants receiving intercurrent antibiotics, while green lines refer to participants not receiving intercurrent antibiotics.
(PDF)

**S6 Fig.** Principal coordinates analysis plot with Procrustes transformation of OW samples using (A and C) Bray-Curtis and (B and D) Jaccard coloured by bronchoscopy number and intercurrent antibiotic use with and without lines. M^2: The summed squares of deviations, First: first bronchoscopy, Second: Second bronchoscopy. A line was drawn between samples from the same participant. Red lines refer to participants receiving intercurrent antibiotics, while green lines refer to participants not receiving intercurrent antibiotics.
(PDF)

**S7 Fig.** Principal coordinates analysis plot of BAL samples using (A and C) Bray-Curtis and (B and D) Jaccard coloured by bronchoscopy number and intercurrent antibiotic use with and without lines. First: first bronchoscopy, Second: Second bronchoscopy. A line was drawn between samples from the same participant. Red lines refer to participants receiving intercurrent antibiotics, while green lines refer to participants not receiving intercurrent antibiotics.
(PDF)

**S8 Fig.** Principal coordinates analysis plot with Procrustes transformation of BAL samples using (A and C) Bray-Curtis and (B and D) Jaccard coloured by bronchoscopy number and intercurrent antibiotic use with and without lines. M^2: The summed squares of deviations, First: first bronchoscopy, Second: Second bronchoscopy. A line was drawn between samples from the same participant. Red lines refer to participants receiving intercurrent antibiotics, while green lines refer to participants not receiving intercurrent antibiotics.
(PDF)

## Acknowledgments

The MicroCOPD is a large study with many co-workers. The authors wish to give their thanks to Christine Drengenes, Per Sigvald Bakke, Øistein Svanes, Sverre Lehmann, Marit Elisabet Aardal, Tuyen Thi Van Hoang, Tharmini Kalananthan, Randi Sandvik, Eli Nordeide, Hildegunn Bakke Fleten, Tove Folkestad, Ane Aamli Gagnat, Kristina Apalseth, Stine Lillebø, and Lise Østgård Monsen (Haukeland University Hospital and University of Bergen).

## Author Contributions

**Conceptualization:** Einar M. H. Martinsen, Tomas M. L. Eagan, Rune Nielsen.

**Data curation:** Einar M. H. Martinsen, Tomas M. L. Eagan, Elise O. Leiten, Rune Nielsen.

**Formal analysis:** Einar M. H. Martinsen, Andreu Paytuví-Gallart.

**Funding acquisition:** Tomas M. L. Eagan, Rune Nielsen.

**Investigation:** Einar M. H. Martinsen, Tomas M. L. Eagan, Harald G. Wiker, Elise O. Leiten, Gunnar R. Husebø, Kristel S. Knudsen, Solveig Tangedal, Walter Sanseverino, Andreu Paytuví-Gallart, Rune Nielsen.

**Methodology:** Einar M. H. Martinsen, Tomas M. L. Eagan, Rune Nielsen.

**Project administration:** Tomas M. L. Eagan, Rune Nielsen.

**Software:** Einar M. H. Martinsen, Andreu Paytuví-Gallart.

**Supervision:** Tomas M. L. Eagan, Rune Nielsen.

**Writing – original draft:** Einar M. H. Martinsen.

**Writing – review & editing:** Einar M. H. Martinsen, Tomas M. L. Eagan, Harald G. Wiker, Elise O. Leiten, Gunnar R. Husebø, Kristel S. Knudsen, Solveig Tangedal, Walter Sanseverino, Andreu Paytuví-Gallart, Rune Nielsen.

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
