## [Decision Letter · Decision Letter 0]

13 Oct 2021

PONE-D-21-19426A longitudinal study of the pulmonary mycobiome in subjects with and without chronic obstructive pulmonary diseasePLOS ONE

Dear Dr. Martinsen,

Thank you for submitting your manuscript to PLOS ONE. After careful consideration, we feel that it has merit but does not fully meet PLOS ONE’s publication criteria as it currently stands. Therefore, we invite you to submit a revised version of the manuscript that addresses the points raised during the review process.

In addition to an external reviewer, I have also provided an extensive review of this manuscript myself. Please review all comments and suggestions from the Reviewers.

We look forward to receiving your revised manuscript.

Kind regards,

Christine M. Freeman

Academic Editor

PLOS ONE

Journal Requirements:

2. Please provide additional details regarding participant consent. In the ethics statement in the Methods and online submission information, please ensure that you have specified whether consent was informed."

3. Thank you for stating the following in the Competing Interests/Financial Disclosure* (delete as necessary) section:

“Einar M. H. Martinsen, Elise O. Leiten, Gunnar Husebø, and Solveig Tangedal declare no conflict of interest. Walter Sanseverino and Andreu Paytuví-Gallart are employed at Sequentia Biotech SL. Tomas M. L. Eagan reports lecture fees from Boehringer and AstraZeneca, and grants from GlaxoSmithKline outside the submitted work. Harald G. Wiker reports being head of the educational programme for medicine at the University of Bergen. Kristel S. Knudsen reports lecture fees from Boehringer Ingelheim and Roche. Rune Nielsen reports grants from Boehringer Ingelheim, GlaxoSmithKline, AstraZeneca, and the Timber Merchant Delphins Endowment, in addition to lecture fees from GlaxoSmithKline and AstraZeneca. Rune Nielsen also reports being unpaid member of The Norwegian Respiratory Society, unpaid GOLD national delegate, unpaid ERS national delegate, and paid member of the Reference group for new Norwegian guidelines for COPD (The Norwegian Directorate of Health). This does not alter our adherence to PLOS ONE policies on sharing data and materials.”

We note that one or more of the authors are employed by a commercial company: GlaxoSmithKline,Boehringer and AstraZeneca,

4. We noted in your submission details that a portion of your manuscript may have been presented or published elsewhere. Please clarify whether this [conference proceeding or publication] was peer-reviewed and formally published. If this work was previously peer-reviewed and published, in the cover letter please provide the reason that this work does not constitute dual publication and should be included in the current manuscript.

5. We note that you are reporting an analysis of a microarray, next-generation sequencing, or deep sequencing data set. PLOS requires that authors comply with field-specific standards for preparation, recording, and deposition of data in repositories appropriate to their field. Please upload these data to a stable, public repository (such as ArrayExpress, Gene Expression Omnibus (GEO), DNA Data Bank of Japan (DDBJ), NCBI GenBank, NCBI Sequence Read Archive, or EMBL Nucleotide Sequence Database (ENA)). In your revised cover letter, please provide the relevant accession numbers that may be used to access these data. For a full list of recommended repositories, see http://journals.plos.org/plosone/s/data-availability#loc-omics or http://journals.plos.org/plosone/s/data-availability#loc-sequencing.

Additional Editor Comments (if provided):

Please address all comments from the Reviewers.

Reviewers' comments:

Reviewer's Responses to Questions

**Comments to the Author**

1. Is the manuscript technically sound, and do the data support the conclusions?

Reviewer #1: Yes

Reviewer #2: Yes

2. Has the statistical analysis been performed appropriately and rigorously? 

Reviewer #1: Yes

Reviewer #2: Yes

3. Have the authors made all data underlying the findings in their manuscript fully available?

Reviewer #1: Yes

Reviewer #2: Yes

4. Is the manuscript presented in an intelligible fashion and written in standard English?

Reviewer #1: Yes

Reviewer #2: Yes

5. Review Comments to the Author

Reviewer #1: This is a very well written manuscript that reports valuable information relative to the temporal stability of the human lung mycobiome.

I have only a few suggestions for improvement:

Lines 213 and 495. Change “corresponds” to “correspond”

Legends for figures 3 and 4A. I suggest adding to the legend information regarding the significance of the statistical values that are shown on the brackets on the figures.

I suggest an expanded explanation of intercurrent antibiotic use in Materials and Methods. This section currently states that recent antibiotic use was a reason for postponing participation in the study. But the fact that some participants then had antibiotic treatment becomes a variable in the study with no real discussion of the antibiotics involved or the timeline of their use.

Line 318. Change “was” to “were”

Line 320. Perhaps change “paper” to “study”

Line 321. Perhaps change “but were” to “but it was”

There is a recent paper by Rubio-Portillo et al. that compares the lung mycobiome to the fungal community found in the home environment. The authors might want to see if this study is relevant to the work presented in the current manuscript. [Rubio-Portillo et al. Microorganisms 8, 1717 (2020)].

Reviewer #2: The goal of this study is to analyze the stability of the mycobiome in BAL and oral wash over time in both healthy participants and participants without COPD. Repeat bronchoscopies were performed between 3 and 12 months. Mycobiome data from the oral wash and broncholaveolar lavage have been published, but this study is focusing on the stability of the mycobiome and therefore has a different scientific question and purpose.

The statistical analysis is appropriate and the data support the conclusions. The manuscript is clearly written.

Comments:

1. Based on the title and the abstract which states that "no apparent effect was seen on taxonomy from intercurrent antibiotic use or participant category", I expected to see analyses pertaining to the COPD status of the participant. I see that Figure 2 is separated into participant category, but where are the statistics that support the finding that there are no differences. I did not see this in the results.

2.Page 5, Line 116 states that "the second portion of the BAL was utilised for mycobiome analysis". Is there any reason to think that this could impact the results? Would the signal from the first portion have been more robust?

3. Table 1. Please include statistics for all demographics. This information is useful in evaluating the data.

4. Page 15, line 324. This information should be included in the results section prior to Figure 3. Please explain why there are almost 50 samples in Figure 2 and then only a handful of samples in Figure 3. All figure legends should include the number of participants or samples.

5. Did you consider evaluating inhaled corticosteroid usage or current vs former smoking status to see whether these variables had any influence on the mycobiome stability?

6. Page 12, line 258 -259: Please indicated that you are discussing S6B

6. PLOS authors have the option to publish the peer review history of their article (what does this mean?). If published, this will include your full peer review and any attached files.

Reviewer #1: No

Reviewer #2: Yes. **Christine M. Freeman**

---

## [Author Response · Author response to Decision Letter 0]

20 Nov 2021

Reviewer 1

C10 This is a very well written manuscript that reports valuable information relative to the temporal stability of the human lung mycobiome.

I have only a few suggestions for improvement:

Lines 213 and 495. Change “corresponds” to “correspond”

A10 Thank you for making us aware of this misspelling. We have changed as suggested in the revised manuscripts (line 216 and 509 in revised manuscript). 

C11 Legends for figures 3 and 4A. I suggest adding to the legend information regarding the significance of the statistical values that are shown on the brackets on the figures.

A11 We have now added additional information of the chosen significance level. 

C12 I suggest an expanded explanation of intercurrent antibiotic use in Materials and Methods. This section currently states that recent antibiotic use was a reason for postponing participation in the study. But the fact that some participants then had antibiotic treatment becomes a variable in the study with no real discussion of the antibiotics involved or the timeline of their use.

A12 We appreciate the suggestion and have added more information on intercurrent antibiotic use in the Materials and methods section (line 86 and 95-98 in revised manuscript). 

C13 Line 318. Change “was” to “were”.

A13 Thank you for making us aware of this misspelling. We have changed as suggested in the revised manuscript.

C14 Line 320. Perhaps change “paper” to “study”

A14 We believe it is more correct to use “paper” in this case as we have published several findings from the study while the observation mentioned in this sentence is specific for this paper. However, we added “fungal” in front of alpha diversity for clarity (line 327 in revised manuscript).

C15 Line 321. Perhaps change “but were” to “but it was”

A15 We agree. Changed as suggested in the revised manuscript.

C16 There is a recent paper by Rubio-Portillo et al. that compares the lung mycobiome to the fungal community found in the home environment. The authors might want to see if this study is relevant to the work presented in the current manuscript. [Rubio-Portillo et al. Microorganisms 8, 1717 (2020)].

A16 We would like to thank the reviewer for the suggested revisions. The suggested paper examines an interesting association and add important data on the research of the lower respiratory tract mycobiome. We believe, however, that the study’s relevance is not directly comparable to our paper. There are several studies investigating the lower respiratory mycobiome, but few looking at the mycobiomes stability which is the aim in the current paper. We have chosen not to include the suggested paper as our paper specifically examine mycobiome stability but are grateful for an interesting read!

Reviewer 2

C17 The goal of this study is to analyze the stability of the mycobiome in BAL and oral wash over time in both healthy participants and participants without COPD. Repeat bronchoscopies were performed between 3 and 12 months. Mycobiome data from the oral wash and broncholaveolar lavage have been published, but this study is focusing on the stability of the mycobiome and therefore has a different scientific question and purpose.

The statistical analysis is appropriate and the data support the conclusions. The manuscript is clearly written.

Based on the title and the abstract which states that "no apparent effect was seen on taxonomy from intercurrent antibiotic use or participant category", I expected to see analyses pertaining to the COPD status of the participant. I see that Figure 2 is separated into participant category, but where are the statistics that support the finding that there are no differences. I did not see this in the results.

A17 We agree that information in the abstract could be misleading. There are no analyses on the differences in the healthy lung mycobiome and the COPD mycobiome in the current paper as this was the focus of our previous mycobiome paper [1]. It is possible to visually examine differences by time in the different participant groups’ mycobiomes by looking at Fig 2, but the reviewer is right that no statistics were performed to support this. We have specified by adding “visually” in front of “seen” in our statements (in abstract and line 234 and 313 in revised manuscript).

The title refers to the inclusion and examination of longitudinal data from both subjects with and without COPD, not the comparison between these groups. Thus, we believe the title is descriptive.

C18 Page 5, Line 116 states that "the second portion of the BAL was utilised for mycobiome analysis". Is there any reason to think that this could impact the results? Would the signal from the first portion have been more robust?

A18 We appreciate input on this issue. As stated, only the second fraction of the BAL was used in the mycobiome analyses in the MicroCOPD study. Another paper by our research group has shown high similarity between the first and the second fraction of BAL, suggesting little impact from choice of fraction [2]. Furthermore, at least in theory, use of a second fraction could be a method to reduce impact from potential bronchoscopic carryover [3]. We have added some information on this in the discussion part of the manuscript (line 369-371 in revised manuscript).

C19 Table 1. Please include statistics for all demographics. This information is useful in evaluating the data.

A19 We agree, and have included this in the revised manuscript.

C20 Page 15, line 324. This information should be included in the results section prior to Figure 3. Please explain why there are almost 50 samples in Figure 2 and then only a handful of samples in Figure 3. All figure legends should include the number of participants or samples.

A20 Abundance data in a mycobiome study is not directly comparable. Each sample has its own sequencing depth or sum of abundances. Several techniques have been proposed to deal with this compositionality, including rarefaction (subsampling without replacement to a given sampling depth). We described the rarefaction process in the Materials and methods section (line 161 and 162 in revised manuscript). In short, the rarefaction subsamples each sample to a given rarefaction or sampling/sequencing depth without replacement. Samples with a sequence count below this value will be discarded. The sampling depth was set as high as possible while excluding a minimum of samples. Still, rarefaction omitted most BAL samples in the longitudinal paper (paper IV), leaving only 12 left for analyses and decreasing power in the diversity analyses including BAL samples. We admit that rarefaction normalise data at the expense of power loss. However, we used QIIME 2 [4], one of the main pipelines used to perform microbiome analyses, and their diversity metric plugin or command has an included rarefaction step.

 We agree that this information is better suited for the Results section and it has been moved in the revised manuscript (to line 249-250 in revised manuscript). We also reformulated the Discussion section to clarify on the issue (line 331-333 in revised manuscript).

 Figure legends for Fig 2 and Fig 3 are updated with information on sample sizes. The information was included in the plot for Fig 4 and no changes were done to the figure legend.

C20 Did you consider evaluating inhaled corticosteroid usage or current vs former smoking status to see whether these variables had any influence on the mycobiome stability?

A20 This is an interesting suggestion and also something we considered. However, due to the low number of samples such sub-analyses seemed unfit, for instance due to the already mentioned rarefaction process. Furthermore, we have previously shown that the pulmonary mycobiomes do not seem to be influenced by smoking habits or use of inhaled steroids [1]. We included intercurrent antibiotics due to a potential influence on the stability through possible interactions with the bacterial microbiome. Further studies with a larger sample size are definitely warranted to examine effects on the mycobiome stability from both smoking and use of inhaled steroids.

C21 Page 12, line 258 -259: Please indicated that you are discussing S6B.

A21 Thanks for suggestion. Clarified in revised manuscript.

We hope that you find the revision satisfactory, but are of course happy to provide a second revision if necessary. All changes are marked with “track changes” function in the revised manuscript. 

We thank the reviewers and the editor for helping improve our paper. Please do not hesitate to contact the corresponding author if unresolved issues remain.

---

## [Decision Letter · Decision Letter 1]

5 Apr 2022

A longitudinal study of the pulmonary mycobiome in subjects with and without chronic obstructive pulmonary disease

PONE-D-21-19426R1

Dear Dr. Martinsen,

We’re pleased to inform you that your manuscript has been judged scientifically suitable for publication and will be formally accepted for publication once it meets all outstanding technical requirements.

Kind regards,

Christine M. Freeman

Academic Editor

PLOS ONE

Additional Editor Comments (optional):

Reviewers' comments:

Reviewer's Responses to Questions

**Comments to the Author**

1. If the authors have adequately addressed your comments raised in a previous round of review and you feel that this manuscript is now acceptable for publication, you may indicate that here to bypass the “Comments to the Author” section, enter your conflict of interest statement in the “Confidential to Editor” section, and submit your "Accept" recommendation.

Reviewer #1: All comments have been addressed

Reviewer #2: All comments have been addressed

2. Is the manuscript technically sound, and do the data support the conclusions?

Reviewer #1: Yes

Reviewer #2: Yes

3. Has the statistical analysis been performed appropriately and rigorously? 

Reviewer #1: Yes

Reviewer #2: Yes

4. Have the authors made all data underlying the findings in their manuscript fully available?

Reviewer #1: Yes

Reviewer #2: Yes

5. Is the manuscript presented in an intelligible fashion and written in standard English?

Reviewer #1: Yes

Reviewer #2: Yes

6. Review Comments to the Author

Reviewer #1: (No Response)

Reviewer #2: (No Response)

7. PLOS authors have the option to publish the peer review history of their article (what does this mean?). If published, this will include your full peer review and any attached files.

Reviewer #1: No

Reviewer #2: **Yes: **Christine Freeman

---

## [Editor Report · Acceptance letter]

13 Apr 2022

PONE-D-21-19426R1 

A longitudinal study of the pulmonary mycobiome in subjects with and without chronic obstructive pulmonary disease 

Dear Dr. Martinsen:

I'm pleased to inform you that your manuscript has been deemed suitable for publication in PLOS ONE. Congratulations! Your manuscript is now with our production department. 

Kind regards, 

on behalf of

Dr. Christine M. Freeman 

Academic Editor

PLOS ONE